# Competing itinerant and local spin interactions in kagome metal FeGe

Lebing Chen[1], Xiaokun Teng[1], Hengxin Tan[2], Barry L. Winn[3], Garrett E. Granroth[3], Feng Ye[3], D. H. Yu[4], R. A. Mole[4], Bin Gao[1], Binghai Yan[2], Ming Yi[1] & Pengcheng Dai[1] ✉

The combination of a geometrically frustrated lattice, and similar energy scales between degrees of freedom endows two-dimensional Kagome metals with a rich array of quantum phases and renders them ideal for studying strong electron correlations and band topology. The Kagome metal, FeGe is a noted example of this, exhibiting A-type collinear antiferromagnetic (AFM) order at $T_N \approx 400$ K, then establishes a charge density wave (CDW) phase coupled with AFM ordered moment below $T_{CDW} \approx 110$ K, and finally forms a $c$-axis double cone AFM structure around $T_{Canting} \approx 60$ K. Here we use neutron scattering to demonstrate the presence of gapless incommensurate spin excitations associated with the double cone AFM structure of FeGe at temperatures well above $T_{Canting}$ and $T_{CDW}$ that merge into gapped commensurate spin waves from the A-type AFM order. Commensurate spin waves follow the Bose factor and fit the Heisenberg Hamiltonian, while the incommensurate spin excitations, emerging below $T_N$ where AFM order is commensurate, start to deviate from the Bose factor around $T_{CDW}$, and peaks at $T_{Canting}$. This is consistent with a critical scattering of a second order magnetic phase transition with decreasing temperature. By comparing these results with density functional theory calculations, we conclude that the incommensurate magnetic structure arises from the nested Fermi surfaces of itinerant electrons and the formation of a spin density wave order.

Materials with flat electronic bands near the Fermi level are interesting because they display a wide range of novel phenomena, such as unconventional superconductivity[1,2], nematicity[3], strange metallicity[4], generalized Wigner crystal state[5], fractional Chern insulator states[6], time reversal symmetry breaking charge order[7], and exotic magnetism[8]. This arises because a system exhibiting a large density of states near the Fermi level can respond to instabilities under different types of interaction when the Coulomb repulsive energy is on the same order as the electronic kinetic energy, giving rise to exotic properties due to electron correlations. While flat electronic bands near the Fermi level can be achieved through magic-

angle twisted bilayer graphene[1], flat electronic bands can also naturally occur in metals with two-dimensional (2D) kagome lattice structure from destructive interference of electronic hopping pathways around the kagome bracket[9–11]. For this reason, there is much interest in studying metals with kagome lattice structure[12–15]. For weakly electron correlated kagome metals such as $AV_3Sb_5$ ($A$ = Cs, Rb, K), where electronic structures can be well-described by density functional theory (DFT) and flat electronic bands are far away from the Fermi level, there are coexisting charge density wave (CDW) and superconductivity without long-range magnetic order[16–21]. For electron-correlated kagome metals such as the FeSn family, where

[1]Department of Physics and Astronomy, Rice University, Houston, TX 77005, USA. [2]Department of Condensed Matter Physics, Weizmann Institute of Science, Rehovot 7610001, Israel. [3]Neutron Scattering Division, Oak Ridge National Laboratory, Oak Ridge, TN 37831, USA. [4]Australian Nuclear Science and Technology Organisation, Lucas Heights, NSW 2234, Australia. ✉e-mail: pdai@rice.edu

electronic structures can only be approximately described by re-normalized DFT calculations[22], there is long-range magnetic order but without CDW and superconductivity[12,22–25]. Recently, FeGe, iso-electronic to FeSn[26–31], was found to have CDW order deep inside the antiferromagnetic (AFM) ordered phase that couples with a magnetic ordered moment[32,33]. FeGe is the only known magnetic kagome system to develop CDW order. By comparing the temperature dependence of electronic structures measured by angle-resolved photoemission spectroscopy (ARPES) with DFT calculations, it was found that FeGe is a moderately electron-correlated magnet where the density of states near the Fermi level is dominated by Fe $3d$ orbitals. Furthermore, DFT calculations suggest that the geometrically frustrated flat bands are near the Fermi level in the high-temperature paramagnetic state, and are spin-split in the AFM phase, out of which the CDW order is observed to develop[34]. Therefore, it is interesting to study the potential connection between electronic structure and magnetism in FeGe.

At the Néel temperature $T_N \approx 400$ K, FeGe exhibits A-type AFM order with $c$-axis polarized moments in alternating ferromagnetic (FM) kagome layers (Fig. 1a)[30,31]. Then at $T_{CDW} \approx 110$ K, a $2 \times 2 \times 2$ CDW phase occurs that enhances the ordered magnetic moments[32–35]. Finally, below $T_{Canting} \approx 60$ K, incommensurate magnetic peaks appear around magnetic Bragg peaks along the $c$-axis at $q_{IC} = (L \pm \delta)$, where $\delta = 0.04$ r.l.u. and $L = \pm 1/2, 3/2, \cdots$, that has been interpreted as evidence for the $c$-axis double-cone AFM structure (Fig. 1b–g)[30–32]. Similar observations are also found in kagome magnets $YbMn_6Ge_{6-x}Sn_x$[36], $YMn_6Sn_6$[37,38], and $YMn_6Ge_6$[39].

In metallic crystalline solids, magnetic order can be described by either a quantum spin model with local moments on each atomic site (Fig. 1a–c)[40,41], or quasiparticle spin-flip excitations between the valence and conduction bands near the Fermi level (termed spin density wave) as the consequence of electron-electron correlations (Fig. 1h–j)[42]. At the long wavelength limit (small momentum transfer $q$), spin waves should be well-defined bosonic modes and are expected to follow the Bose population factor in the magnetic ordered state. In addition, the energy ($E$) and momentum dispersion of spin waves can be fitted by a Heisenberg Hamiltonian with several nearest neighbor (NN) exchange couplings, thus providing direct information on the strength of the itinerant electron induced Ruderman-Kittel-Kasuya-Yosida (RKKY) magnetic interactions[41]. For materials with strong electron correlations such as copper oxide superconductors $La_{2-x}(Ba,Sr)_xCuO_4$[43,44], $YBa_2Cu_3O_{6+x}$[45], and cobalt oxide $La_{2-x}Sr_xCoO_4$[46], spin excitations exhibit hourglass-like dispersions that can be well-described by localized moments in an inhomogeneous spin-charge separated stripe phase[47], although the Fermi surface nesting explanation also cannot be totally ruled out[48]. For intermediate electron correlated materials such as iron pnictides[49], both Fermi surface nesting of itinerant electrons and localized moments contribute to spin excitations[50].

To understand the microscopic origin of incommensurate magnetic order in FeGe, we carried out inelastic neutron scattering experiments to measure temperature and magnetic field dependent incommensurate order and the associated spin excitations. If the spin structure of FeGe follows the local moment picture, the canted

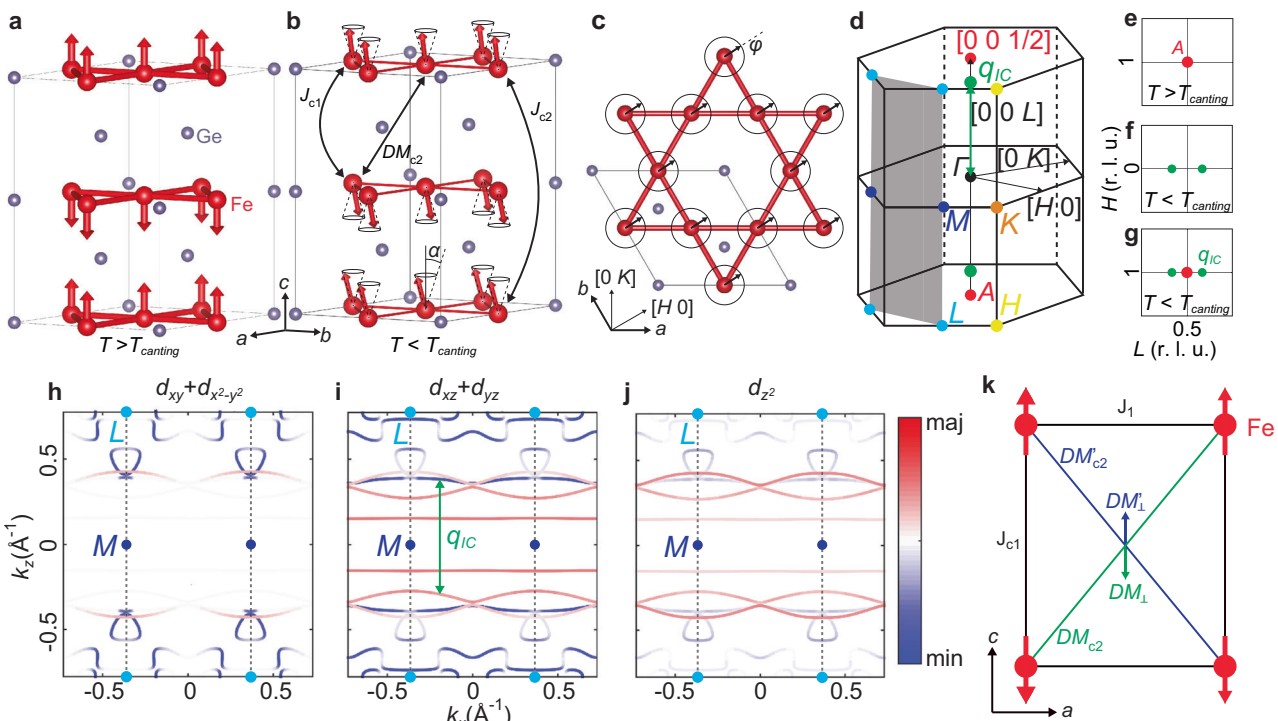

**Fig. 1 | Crystal, magnetic, and electronic structures of FeGe. a** The magnetic unit cell of FeGe in the A-type AFM state, (**b**) The incommensurate double cone AFM structure with a canting angle $\alpha$, showing interlayer nearest neighbor exchange $J_{c1}$, next-nearest neighbor exchange $J_{c2}$, and possible interlayer DM interaction $DM_{c2}$. The spiral spin structure in **b** and **c** are speculations from previous literature[30,31]. **c** The kagome Fe layer in the incommensurate phase with canted spins at an azimuth angle $\phi$. Here the CDW-induced lattice distortion is not pictured. **d** The first Brillouin zone of pristine FeGe (symmetric phase above $T_N$) with high-symmetry points. The positions of incommensurate magnetic Bragg peaks are marked as green dots. The shaded area corresponds to the reciprocal space shown in panels **h**–**j**. All slices and cuts in this work are integrated between [$H, H$] = [−0.03, 0.03] r.l.u., [−$K, K$] = [−0.05, 0.05] r.l.u.. (e-g) Schematics of the neutron magnetic Bragg peak intensity at (**e**) $T > T_{Canting}$ around (1, 0, 0.5), (**f**) $T < T_{Canting}$ around (0, 0, 0.5), and (**g**) $T < T_{Canting}$ around (1, 0, 0.5). **h**–**j** Orbital-selective DFT band structure calculations in the $k_x$-$k_z$ plane denoted by the shaded area in **d**, the respective $M$ and $L$ points are shown using blue and cyan dots, respectively. The nesting wavevector $q_{IC}$ (green double arrow) in **i** corresponds to the incommensurate magnetic Bragg peak position shown in **d**, **f**, **g**. **k** Schematics of the effective DM vector on the A-type AFM spins bonded by $DM_{c2}$, showing zero net contribution.

magnetic structure should be stabilized by the competition between the nearest interlayer interaction $J_{c1}$ and the next-nearest layer $J_{c2}$ along the $c$-axis (Fig. 1b)[31]. On the other hand, incommensurate magnetic peaks could also be spin density wave-like modulations arising from electron-hole Fermi surface nesting at $q = q_{IC}$, analogous to the collinear magnetic order in iron pnictides[50]. Since a double-cone canted AFM structure as observed in FeGe is not supported by a reasonable Heisenberg Hamiltonian with Dzyaloshinskii-Moriya (DM) interactions and magnetic anisotropy within the centrosymmetric kagome lattice structure of FeGe (Fig. 1k)[51], a determination of the microscopic origin of the incommensurate peaks in FeGe will shed new light on our understanding of the magnetic structure and interactions in magnetic kagome lattice materials.

Here we report neutron scattering studies of the magnetic structure and low-energy spin excitations of FeGe as a function of temperature and in-plane magnetic field along the $[H, -H, 0]$ direction. We confirm that an in-plane field of up to 11 T suppresses the incommensurate magnetic elastic scattering at $(0, 0, \pm\delta)$ but keeping the incommensurability $\delta$ unchanged[30,31]. In the canted AFM phase ($T < T_{Canting}$), gapless spin excitations stem from incommensurate wave vectors $q_{IC} = (L \pm \delta)$ and merge with increasing energy into gapped spin waves from A-type AFM order at $L = 0.5$. Surprisingly, incommensurate gapless spin excitations persist to temperatures well above $T_{Canting}$ and $T_{CDW}$, where static AFM order is commensurate, and vanish only around $T_N$. The spin gap at commensurate $L = 0.5$ increases with increasing temperature, contrary to the expectation of spin-orbit coupling induced anisotropy gap but consistent with the increase in the magnitude of $c$-axis magnetic field needed to induce spin-flop transition[30–32,52]. By carefully fitting the overall spin excitation dispersions along the $L$ direction in the A-type and canted AFM phases using the linear spin wave theory (LSWT) within a Heisenberg Hamiltonian at temperatures across $T_{Canting}$[40,41], we find that spin waves can be well described by the NN $c$-axis exchange coupling and the incommensurate magnetic peaks below $T_{Canting}$ cannot arise from the proposed double-cone canted AFM structure[30,31]. Instead, the incommensurate peaks are likely due to Fermi surface nesting, arising from flat electronic bands near the Fermi level around $T_N$. On cooling below $T_{CDW}$, the opening of electronic gaps near Van Hove singularities further modifies the incommensurate peaks, setting up magnetic critical scattering associated with $T_{Canting}$. For comparison, low-energy spin waves from commensurate A-type AFM order can be well understood by a local moment Heisenberg Hamiltonian. Therefore, low-temperature magnetic phases of FeGe arise from competition amongst the local moment exchange, magnetic anisotropy, and spin density wave interactions from Fermi surface nesting, most likely due to flat electronic bands near the Fermi level around $T_N$ and associated electron correlation effects.

## Experimental results

We first consider spin excitations in the commensurate A-type AFM phase at a temperature well above the incommensurate AFM and CDW-ordered phases ($T > T_{CDW} > T_{Canting}$). Figure 2a, c show the overall spin wave spectrum along the $[0, 0, L]$ direction and low-energy spin excitations near $(0,0,0.5)$, respectively, at $T = 120$ K. While the overall spin wave spectrum has a band top of ~22 meV (Fig. 2a), the low-energy excitations reveal two components: a commensurate spin excitation with high intensity gapped around 1 meV, and low-intensity gapless spin excitations centered at $\mathbf{Q} = (0, 0, 0.5 \pm \delta)$, where $\delta = 0.04$ r.l.u. is the ordering wave vectors of incommensurate peaks below $T_{Canting}$ (Fig. 2c, f). The observed spin gap at commensurate wavevector $(0, 0, 0.5)$ in FeGe is the single-ion anisotropy gap, its value of ~1 meV is similar to the anisotropy gap of ~1.5 meV at $(0, 0, 0.5)$ in spin waves of FeSn, where there are no incommensurate spin excitations around $\mathbf{Q} = (0, 0, 0.5 \pm \delta)$[24,25]. Figure 2b shows the overall spin wave

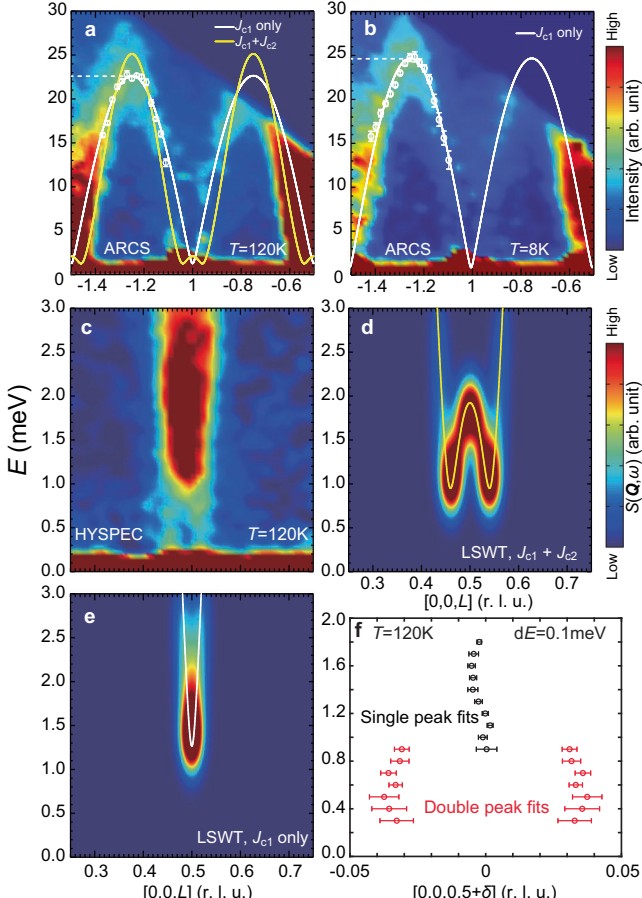

**Fig. 2 | Spin excitation spectrum along the $[0, 0, L]$ direction. a** Overall spin waves along the $[0, 0, L]$ direction at 120 K. The white and yellow lines are the best LSWT fit using $J_{c1}$-only model and the $J_{c1}$-$J_{c2}$ model respectively. The white data points are constant-$Q$ cuts used for fitting. **b** Same spin wave dispersion as (**a**) at 8 K. The band top is ~10% higher than that in **a**. The fitting line assumes the spins are along the $c$-axis. **c** Low-energy spin excitations at 120 K and 0 T, showing a gapped commensurate part and a gapless incommensurate part. **d, e** The low-energy neutron spectra for the $J_{c1}$-$J_{c2}$ model and the $J_{c1}$ only model, respectively. **f** Constant energy fits of the intensity shown in (**c**), with double Gaussian peak fitting at lower energy and single Gaussian fitting at higher energy. The horizontal error bars in **f** are uncertainties of peak positions obtained from single and double peak fits.

spectrum along the $[0, 0, L]$ direction at 8 K, showing slight hardening of the zone boundary magnon.

Since previous neutron diffraction experiments reveal that an in-plane magnetic field can dramatically change the magnetic intensity of incommensurate peaks and modify magnetic structure[30,31], it will be interesting to determine the temperature and in-plane magnetic field dependence of the low-energy spin excitations. Figure 3a, b show $\mathbf{Q} - E$ maps of low-energy spin excitations at 70 K and base (2 K), respectively, with zero applied field. Compared with the 120 K case (Fig. 2c), spin excitations at 70 K (Fig. 3a) and 2 K (Fig. 3b) show similar patterns with gapped commensurate and gapless incommensurate spin excitations. However, the spin gap at commensurate wavevector $L = 0.5$ reduces with decreasing temperature, contrary to the expected behavior of an anisotropy gap. By cutting the gapped excitations along the energy at $L = 0.5 \pm 0.01$, we can avoid incommensurate spin excitations and quantitatively determine the temperature dependence of the commensurate anisotropy gap sizes as shown in data points of Fig. 3d–f. We use the equation $I = I_0 + A \cdot \text{Erfc}\,[(E - E_{gap})/\sigma]/[1 - \exp(-E/k_B T)]$ to fit the energy cuts, where $\text{Erfc}(x)$ is the error function simulating finite instrumental resolution, $E_{gap}$ is estimated gap value, $k_B$ is the Boltzmann constant, and the denominator serves as the Bose population factor.

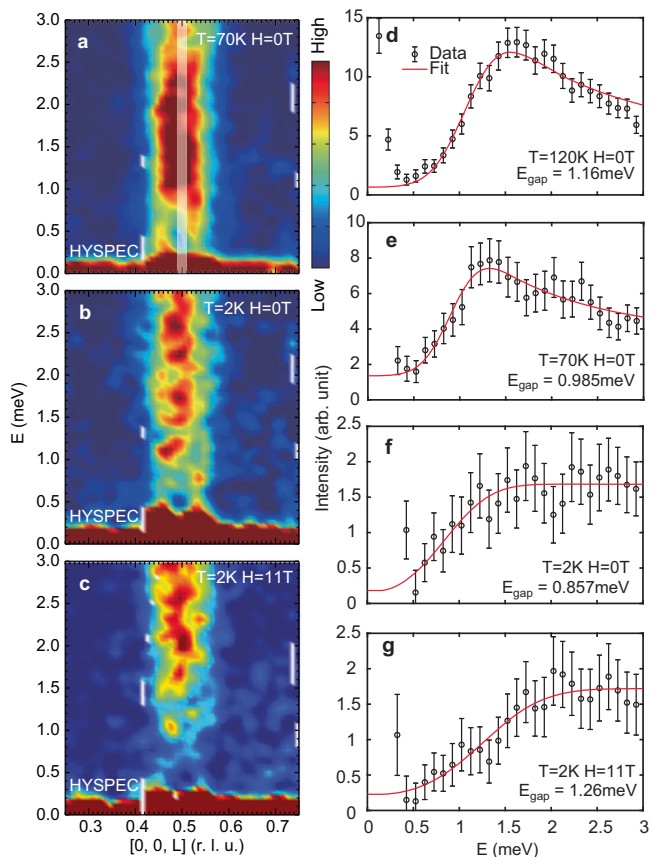

**Fig. 3 | Temperature and in-plane magnetic field dependence of spin excitations. a**–**c** Low-energy spin excitations at (70 K, 0 T), (2 K, 0 T), and (2 K, 11 T), respectively. **d**–**g** the constant-$Q$ ($Q = (0, 0, 0.5 \pm 0.01)$) cuts at (120 K, 0 T), (70 K, 0 T), (2 K, 0 T), and (2 K, 11 T). The vertical error bars in **d**–**g** are statistical errors of 1 standard deviation.

The spin gap values extracted at 120 K, 70 K, and 2 K are $E_{gap} = 1.16 \pm 0.02$, $0.99 \pm 0.03$, and $0.86 \pm 0.06$ meV, respectively. For the 120 K data, we can calculate the single-ion anisotropy $D_z = -0.015$ meV in the $J_{c1}$-only model from the LSWT formula $E_{gap} = 2S\sqrt{2J_{c1}|D_z|}$, where we assume Fe spin $S = 1$. With decreasing temperature, the reduction of the anisotropy gap is comparable with the decrease of the critical $c$-axis aligned magnetic field needed to induce a spin-flop transition[27,32]. Figure 3c shows the impact of an 11-T in-plane magnetic field on the **Q**-$E$ map of Fig. 3b. In addition to suppressing quasi-elastic scattering near the incommensurate wave vectors, the field enhances the spin gap from $E_{gap} = 0.86$ meV at 0-T (Fig. 3f) to 1.26 meV at 11-T (Fig. 3g).

To understand the impact of $T_{Canting}$, $T_{CDW}$, and $T_N$ on the low-energy incommensurate spin excitations, we summarize in Fig. 4 the temperature evolution of the incommensurate spin excitations along the [0, 0, $L$] direction. The incommensurate spin excitations survive up to at least 350 K (Fig. 4a–e), then merge with the commensurate spin waves around $T_N = 400$ K (Fig. 4f, g) as the latter collapse to zero energy. Similar to Fig. 3d–g, we extract the commensurate gap sizes ($E_{gap}$) up to 350 K (Fig. 3h) and find that $E_{gap}$ is proportional to the spin-flop field $H_{SF}$ times the ordered moment $M$ (the red line in Fig. 4i)[32]. This is expected because a spin-flop transition occurs when the Zeeman energy for magnons $g\mu_B H$ exceeds the anisotropy gap energy $E_{gap}$. The temperature dependence of the anisotropy is also consistent with previous torque measurements[29]. The dependence of the magnetic anisotropy on temperature and field modulates the spin wave spectrum by adjusting the spin gap while maintaining both the magnon band structure and the spectral weight.

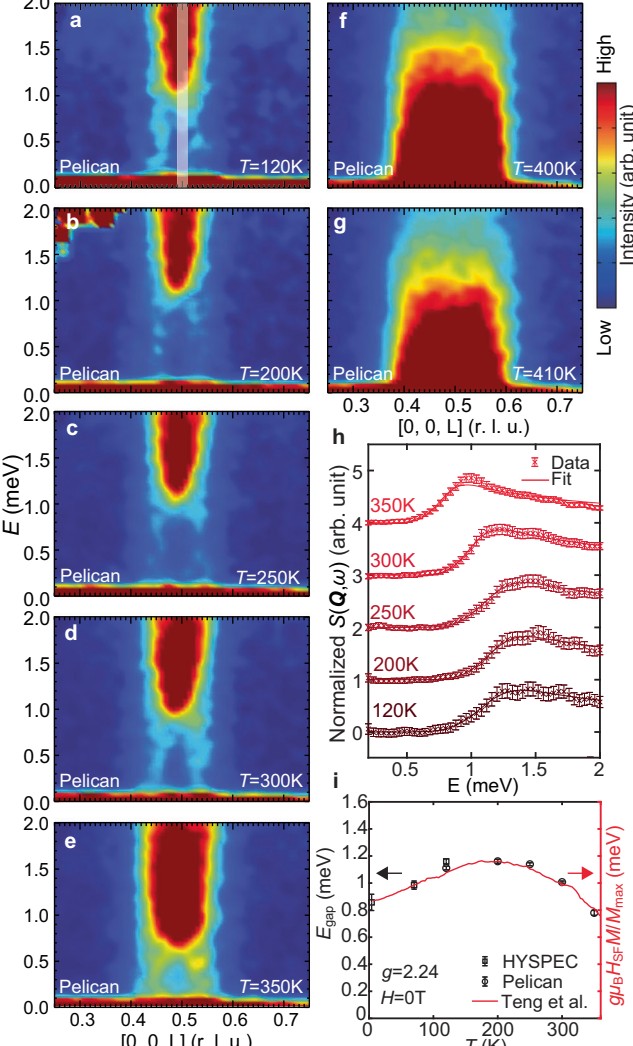

**Fig. 4 | Low-energy spin excitations above $T_{CDW}$. a**–**g** Low-energy spin excitations at 120 K, 200 K, 250 K, 300 K, 350 K, 400 K, and 410 K, respectively. The color bar is scaled with a Bose factor at 1 meV for different temperatures. **h** constant-$Q$ ($Q = (0, 0, 0.5 \pm 0.01)$) cuts from spectra shown in **a**–**e**, with fitting curves specified in the main text. **i** Fitted gap sizes ($E_{gap}$) as a function of temperature, over-plotted with the calculated gap size from the spin-flop field ($H_{SF}$) and ordered magnetic moment ($M$) from[32]. The vertical error bars in **h** are statistical errors of 1 standard deviation.

If both commensurate and incommensurate excitations originate from the same $c$-axis double-cone AFM structure, we would expect both to follow the Bose population factor with increasing temperature, as our muon spin rotation experiments find above 90% magnetic ordered volume fraction below 200 K (unpublished). Figure 5a, b compare the temperature dependence of spin excitations along the [0, 0, $L$] direction at different energies. While $E = 1.5$ meV excitations at the commensurate position follow the Bose population factor $I \propto 1/[1 - \exp(-E/k_B T)]$ from 2 K to 120 K (Fig. 5b), $E = 0.6$ meV spin excitations at incommensurate wave vectors first increase in intensity on warming from 2 K to 70 K, and then decrease intensity from 70 K to 120 K (Fig. 5a). In addition, an 11-T in-plane magnetic field dramatically suppresses the incommensurate magnetic Bragg peaks (Fig. 5c) and reduces incommensurate spin excitations (Fig. 5d), but has limited impact for commensurate spin excitations at $E = 1.5$ meV (Fig. 5d). With increasing temperature from 4 K, the intensity of the incommensurate excitations initially increases, reaching a broad plateau around $T_{Canting}$,

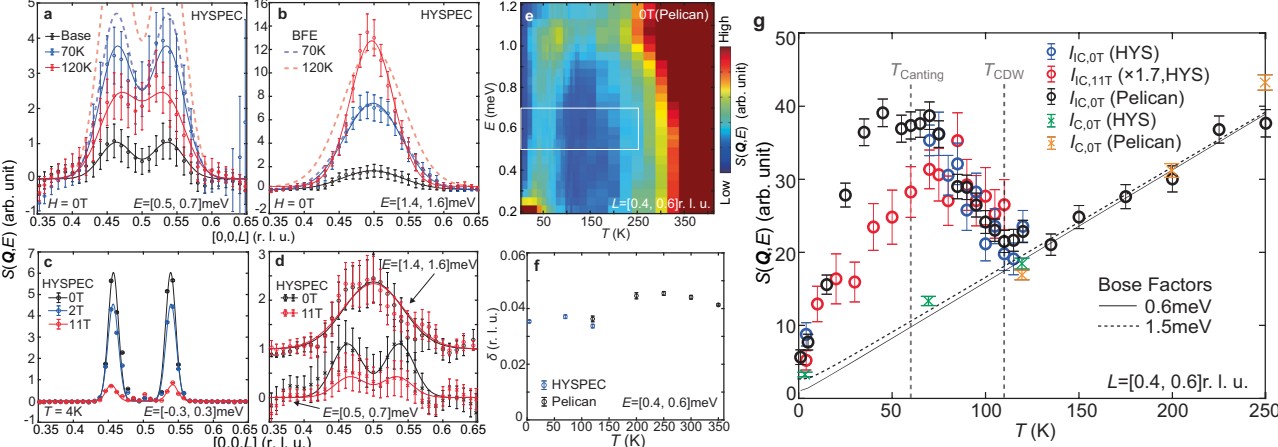

**Fig. 5 | Temperature and field dependence of low-energy spin structure and excitations. a**, **b** Temperature dependence of the incommensurate (0.6 meV) and commensurate (1.5 meV) spin excitations, respectively. The dashed lines in **a**, **b** are the estimated intensities at 70 K and 120 K by multiplying the base temperature intensity with a Bose factor. **c** In-plane field dependence of the incommensurate magnetic Bragg peaks at 4 K (base temperature). **d** In-plane field dependence of the incommensurate and commensurate excitations at base temperature. The vertical error bars in **a**–**d** are propagation errors obtained by subtracting the background scattering. **e** Temperature and energy dependencies of the neutron intensity at $L = [0.4, 0.6]$. The lower intensity above 1.1 meV is a result of limited detector coverage. The white box shows the integration and plot range for **g**. **f** Temperature dependence of the incommensurability $\delta$ at $E = 0.5$ meV. **g** Temperature dependence of the 0.6m meV (circles) and 1.5 meV(crosses) spin excitations under 0T and 11T in-plane field. The black solid and dashed lines show the Bose factor at 0.6 meV and 1.5 meV, respectively. The gray vertical dashed lines mark $T_{\text{Canting}}$ and $T_{\text{CDW}}$. The vertical error bars in **g** are statistical errors of 1 standard deviation.

then subsequently decreases but does not disappear completely (Fig. 5e, g). The temperature range of the plateau between 35 to 75 K indicates a crossover region with physical processes that are not fully understood[31]. The incommensurability $\delta$ is weakly temperature dependent from 4 K to 350 K (Fig. 5f). Figure 5g compares temperature dependence of the incommensurate and commensurate spin excitations at 0 and 11-T in-plane field. With increasing temperature, incommensurate spin excitations at 0.6 meV show a broad peak around $T_{\text{Canting}}$ for both 0 and 11-T in-plane field (open circles in Fig. 5g). More importantly, this critical scattering-like peak has a clear kink at $T_{\text{CDW}}$, and follows the Bose factor for temperatures up to $T = 250$ K. This indicates that the CDW phase transition plays an important role in the formation of the eventual static incommensurate order. In contrast, the commensurate spin wave intensity at 1.5 meV generally follows the Bose factor throughout the temperature range of interest (green crosses in Fig. 5g), consistent with the spin wave picture since the $[0, 0, L]$ dispersion does not change dramatically with temperature (Fig. 2a, b). This discrepant temperature dependence suggests that these two spin excitations come from different origins.

## Discussion

From previous experiments and calculations on the electronic and magnetic structures of FeGe[34], the process of AFM phase transition at $T_N$ can be thought of as follows. At some temperatures above $T_N$, the paramagnetic flat bands split into spin-majority and spin-minority bands, which localizes magnetic moments with interplane AFM couplings from the direct exchange and/or the RKKY interactions. These interplane interactions between localized spins stabilize the A-type AFM magnetic order below $T_N$. However, this picture fails to explain the incommensurate phase in FeGe. In the local moment picture, the double cone AFM structure can arise from competition between the $c$-axis magnetic exchange and single-ion anisotropy energies[31]. Assuming that the centrosymmetric kagome lattice symmetry of a pristine FeGe is preserved below $T_N$, the DM interactions between the interlayer Fe atoms should cancel each other and have zero effect on the spin excitations (Fig. 1k)[51]. Therefore, spin waves along the $[0, 0, L]$ direction in this temperature regime should allow an accurate determination of the NN ($J_{c1}$) and next-nearest neighbor (NNN) layer ($J_{c2}$) magnetic exchange couplings along the $c$-axis (Fig. 1b) using LSWT. To

understand spin waves of FeGe using local exchange interactions, we consider a Heisenberg Hamiltonian

$$H_0 = \sum_{\langle i,j \rangle} J_{ij} \mathbf{S}_i \cdot \mathbf{S}_j + \sum_i D_z (S_i^z)^2, \qquad (1)$$

where $J_{ij}$ indicates magnetic exchange interaction between $i$th and $j$th Fe atoms, $\mathbf{S}_i$ ($\mathbf{S}_j$) is the local spin at $i$ ($j$) site, and $D_z$ stands for single-ion magnetic anisotropy. Since the FM in-plane spin exchange couplings[34] have no effect on spin wave dispersion along the $c$-axis ($L$ direction in reciprocal space), we fit the $c$-axis spin wave dispersion with out-of-plane magnetic exchange couplings and single-ion magnetic anisotropy. Within the local exchange picture, if $J_{c1}$ and $J_{c2}$ are both AFM and satisfy $J_{c1}/J_{c2} = -4\cos(2\pi q_{IC}) = 3.874$, it is possible to have a double cone (canted) AFM structure when exchange energy reduction in the canted phase overcomes the magnetic anisotropy energy[31]. For the canting angle $\alpha < 90°$ in the double cone AFM structure (Fig. 1b), one also needs to consider higher-order magnetic anisotropy terms[31]. Assuming that the incommensurate peaks arise from this $J_{c1} - J_{c2}$ relation, one can fit the spin wave spectra in Fig. 2b, c using LSWT[41]. Compared to pure $J_{c1}$ fits with $J_{c2} = 0$ (white solid line in Fig. 2a), the $J_{c1} - J_{c2}$ model is worse in reproducing both the overall spin wave spectrum as well as its low-energy part (Table 1, yellow solid line in Fig. 2a, d). Both the dispersion and intensity of the low-energy incommensurate spin excitations in Fig. 2c are not compatible with the $J_{c1} - J_{c2}$ model, indicating that the local moment picture is not the underlying mechanism for the canted phase transition below $T_{\text{Canting}}$. LSWT fits to spin wave dispersion at 8 K reveal similar behavior (white solid line in Fig. 2b). Note that $J_{c1}$ and $J_{c2}$ in previous reports are

**Table 1 | Fitting parameters and coefficient of determination $r^2$ for the LSWT fitting on [0 0 L] spin wave data at 120 K using the two models mentioned in the text**

| Model | $J_{c1}$ (meV) | $J_{c2}$ (meV) | $D_z$ (meV) | $r^2$ |
|---|---|---|---|---|
| $J_{c1}$-only | 11.3 ± 0.4 | 0 | −0.015 ± 0.001 | 6.9 |
| $J_{c1} - J_{c2}$ | 25.9 ± 2.7 | 6.7 ± 0.7 | −0.018 ± 0.002 | 60.5 |

The fitting parameter is used to generate the calculation results in Figs. 2a, d, and e.

estimated to be 3.5 meV and 0.9 meV, respectively[31]. These values are dramatically different from Heisenberg fits to the $c$-axis dispersion shown in Fig. 2a.

In the above discussion, we assumed that the inversion symmetries along the $c$-axis in the crystal structure of FeGe are preserved below $T_{CDW}$ (Fig. 1k), and therefore there is no net contribution of DM interactions to the double cone magnetic structure[51]. However, recent X-ray diffraction experiments[35] indicate that the Fe atoms form charge dimers along the $c$-axis as well as moving in the $ab$-plane in the CDW phase. This induces asymmetry in the Fe local environment by introducing unequal bond lengths with its upper and lower neighbors, and will presumably change the interlayer exchange coupling and the magnetic anisotropy (Fig. 1b, and Extended Fig. S1). Due to the distance change of the diagonal bonds between interlayer Fe atoms (Fig. 1k), their respective $DM_{c2}$ interactions are not in balance with each other and will provide a non-zero net contribution to the spin Hamiltonian. In addition, the symmetry breaking induced by the CDW phase may introduce odd-parity magnetic anisotropy terms into the system, as suggested by a precursory enhancement of magnetic susceptibility just before the spin-flop transition below $T_{CDW}$ with a $c$-axis magnetic field[32]. This additional magnetic anisotropy brought by the CDW makes it possible to achieve a canting phase with the canting angle $\alpha < 90°$. Nevertheless, since the precise crystalline lattice structure below $T_{CDW}$ is unknown, it is difficult to determine the impact of CDW order on the incommensurate magnetic scattering below $T_{Canting}$.

However, regardless of the role of CDW order on the incommensurate magnetic order, it cannot be the origin of the incommensurability, as incommensurate spin excitations associated with the eventual static magnetic order below $T_{Canting}$ are present at temperatures well above $T_{CDW}$ of ~ 110 K (Figs. 2–5). These results suggest that the origin of incommensurate magnetic order has no direct connection to CDW phase-associated lattice distortion and DM interactions which, in this case, can only serve for tuning the canting angle[51]. Since the intensity and dispersion of the incommensurate spin excitations are not compatible with the gapped spin waves, we conclude that the local moment double cone magnetic structure suggested originally to explain the observed incommensurate order is problematic. Instead, our data suggest that the Fermi surface nesting along the $L$-direction between spin majority and minority bands creates a spin density wave-like order within the commensurate A-type AFM phase analogous to the collinear magnetic order in iron pnictides[50]. To check this possibility, we performed DFT calculations on the $k_y$-$k_z$ plane to extract the nesting susceptibility $\chi(q)$ in the AFM-ordered state, where ferromagnetism within each Fe layer should split the degenerate electronic bands near the Fermi level into the spin-majority and spin-minority electronic bands with different orbital characteristics[42]. Comparing the possible spin-majority/spin-minority pair nesting excitations for the $d_{xy} + d_{x^2-y^2}$ (Fig. 1h), $d_{xz} + d_{yz}$ (Fig. 1i), and $d_{z^2}$ (Fig. 1j) orbitals, we find that the wave vectors of the observed incommensurate spin excitations most likely correspond to the narrow electronic bands with $d_{xz} + d_{yz}$ orbital characters (Fig. 1i).

An advantage of the itinerant picture is that it does not require specific interlayer magnetic or electronic interactions to achieve the incommensurate phase. According to Figs. 1h–j, the nesting susceptibility is mostly enhanced by the in-plane flattish band structure from the kagome geometry, while the out-of-plane electron dispersion only selects the most favorable $q_{IC}$. Specifically, the fact that the IC excitation intensity increases after the CDW phase transition strongly suggests that the IC order is closely related to the Fermi surface nesting between the Van Hove singularities, which the CDW phase transition mostly modifies. For comparison, A-type AFM order in FeGe below $T_N$ is consistent with the local moment Heisenberg Hamiltonian. The property of the combined itinerant and local picture for FeGe makes it

possible for the application to other kagome systems without reconsidering the detailed interatomic magnetic interactions, and can potentially explain the universality of the incommensurate phase in these kagome metals. If the itinerant electron picture is correct, then the incommensurate phase observed in FeGe and related kagome metals are examples of spin density waves originating from the in-plane strong electron correlations but expressed in the interlayer direction possibly involving RKKY interactions. It will be interesting to determine the spin configurations of the incommensurate phase using neutron polarization analysis where the moment direction of the spin density wave can be conclusively determined[53]. Furthermore, one would expect the sizes of the ordered moments themselves can fluctuate, giving rise to longitudinal spin excitations that can be detected by neutron polarization analysis[54]. Our results demonstrate that the incommensurate magnetic phase in FeGe originates neither from the localized exchange interaction nor from the CDW phase transition, but arises from the nested Fermi surfaces of itinerant electrons, possibly involving flat bands near the Fermi level around $T_N$ and associated electron correlation effects[55].

## Methods

### Single crystal growth and the reciprocal lattice

High-quality single crystals of FeGe were grown by the chemical vapor transport method[32,56]. The crystals are typically $2 \times 2 \times 1$ mm³ in size and 15 mg in mass. Pristine FeGe belongs to the hexagonal space group $P6/mmm$ (191) with lattice constant $a = b = 4.99$ Å, $c = 4.05$ Å. The A-type AFM magnetic structure doubles the $c$-axis as shown in Fig. 1a. However, here we still use the chemical lattice structure for the reciprocal lattice vectors. In this notation, the momentum transfer $\mathbf{Q} = H\mathbf{a}^* + K\mathbf{b}^* + L\mathbf{c}^*$ is denoted as $(H, K, L)$ in reciprocal lattice units (r.l.u.) (Fig. 1c, d). The high symmetry points $\Gamma$, $M$, $K$, $A$, $L$, $H$ in the reciprocal space are specified in Fig. 1d.

### Neutron scattering

Inelastic neutron scattering experiments were performed at the ARCS[57] (Fig. 2a, b) and HYSPEC[58] (for all other figures with neutron data) neutron time-of-flight spectrometers at the Spallation Neutron Source (SNS), Oak Ridge National Laboratory (ORNL) on ~ 0.9 grams of single crystal sample aligned in the $[H, H, L]$ scattering plane. Figure S1a, b show the Bragg peaks of the co-aligned sample. The sample mosaicity perpendicular to the $[0, 0, L]$ is 0.92° in full width at half maximum (FWHM). The Laue pattern of every sample is consistent with the hexagonal structure of FeGe (Fig. S1b), and magnetic susceptibility measurements on selective samples show consistent results compared to previous reports[27,32]. The incommensurate magnetic Bragg peak and excitations are resolution limited, indicating that the homogeneity of the composite sample is good. The incident neutron energies for the ARCS and HYSPEC experiments are $E_i = 45$ meV and 9 meV, respectively. Additionally, experiments with the same sample and geometry were carried out at the Pelican spectrometer located in ANSTO, Australia[59]. The elastic line resolution (in full width at half maximum) of the ARCS, HYSPEC, and Pelican experiments are 2.0 meV, 0.33 meV, and 0.13 meV, respectively. The experiments were performed using rotation sample scanning. The neutron data were analyzed and integrated using the DAVE software[60]. To calculate the neutron intensity from LSWT, we utilized the SPINW software package for the magnon dispersion and instrumental resolution convolution[61]. For the HYSPEC experiment, a vertical magnet was used to apply in-plane magnetic fields, and we subtracted all HYSPEC data by an empty magnet scan with no sample in the beam. The Pelican data is also subtracted by background scans with no sample. All background-subtracted neutron data are labeled with unit "$S(\mathbf{Q}, \omega)$", while all unsubtracted data are labeled with unit "Intensity". To emphasize relevant features, all data displayed are smoothed with a level-3 Gouraud shading.

## Incommensurate spin structure and excitations

Figure S2 shows the detailed cuts of magnetic excitations shown in Figs. 2f and 4f of the main text. All the data are integrated according to the range specified in the main text. Figure S3a shows the temperature dependence of the incommensurate magnetic Bragg peaks under an 11-T in-plane field, which is similar to the temperature dependence of the IC Bragg peak at 0T. The temperature dependence of the (0, 0, 0.5) peak mostly follows the CDW temperature dependence at 0T, but is on top of a temperature-independent magnetic background from in-plane moments induced by the 11T field (Fig. S3b). Figure S3c shows the temperature dependence of the imaginary part of the dynamic susceptibility $\chi''(E)$ at the incommensurate position across $T_{Canting}$, where Fig. S3d is the $\chi''$ at 1.5 meV as a function of [0, 0, $L$] at different temperatures. Combined with Fig. 4 in the main text, we further confirm that while the commensurate excitations above the spin gap follow the Bose factor across $T_{Canting}$, the incommensurate excitations go through a peak around 70 K with additional change at $T_{CDW}$ (Fig. 5g). Figure S4 shows spin excitations at 2 K, 70 K, and 120 K under an in-plane field of 2-T, not much different from the 0-T data. Figure S5 shows the overall temperature dependence of the low-energy spin excitations from base to 410 K used for plotting Fig. 4e, f in the main text, with adaptive color bars.

## Density functional theory calculations

DFT calculations were performed with the Vienna ab initio Simulation Package (VASP)[62]. The generalized gradient approximation parameterized by Perdew-Burke-Ernzerhof[63] is used for the electron-electron exchange interaction throughout. The FeGe structure was fully relaxed until the maximal remaining force on atoms is no larger than 1 meV/Å. An energy cutoff of 350 eV is used for the plane wave basis set. $k$-meshes of $12 \times 12 \times 16$ and $12 \times 12 \times 8$ are employed for sampling the Brillouin zones of the FM and AFM phases, respectively. All Fermi-surface-related properties of both the FM and AFM phases are calculated with the tight-binding Hamiltonian obtained from the Wannier 90 software[64] interfaced with VASP, where the Fe $d$ and Ge $p$ orbital are considered. Notice that in Fig. 1h–j, the Fermi surfaces are for the FM phase without spin-orbital coupling. The Lindhard susceptibility for the Fermi surface nesting is calculated and displayed in Fig. S6. Calculations from FM without SOC and AFM with SOC give qualitatively the same results in the band structure and spin susceptibility, known that AFM has a folded band structure and SOC is relatively weak in FeGe. However, the spin susceptibility calculated from the FM structure gives directly the correct nesting vector while that from AFM gives a folded nesting vector, because the AFM structure has a double unit cell along the c axis. Although these two $q$ vectors are physically equivalent, it is more insightful to demonstrate $q_{IC}$ from the FM structure. The nesting susceptibility of the AFM phase with SOC at $q_{IC}$ is at maximum apart from that around the AFM wavevector (Fig. S6d), which supports the nesting picture as the reason for the IC phase.

## The localized spin model for the incommensurate phase

Here we review the localized moment picture by Beckman et al.[27] to understand the incommensurate phase. In the localized spin model, the Hamiltonian consists of Heisenberg exchange and anisotropy. The related terms are:

$$H = \sum_j H_j, \tag{2}$$

$$H_j = J_{c1}(\boldsymbol{S}_j \cdot \boldsymbol{S}_{j+1}) + J_{c2}(\boldsymbol{S}_j \cdot \boldsymbol{S}_{j+2}) + D_z S_{jz}^2. \tag{3}$$

Here $j$ is the atom layer index, $J_{c1}$ and $J_{c2}$ are defined in Fig. 1 of the main text, and $D_z$ is the single-ion anisotropy. To minimize the Hamiltonian,

we first assume the spin structure to be:

$$\begin{aligned} S_{jx} &= S\cos(2\pi j q_{IC})\sin\alpha, \\ S_{jy} &= S\sin(2\pi j q_{IC})\sin\alpha, \\ S_{jz} &= S\cos(j\pi)\cos\alpha. \end{aligned} \tag{4}$$

Then equation (3) turns into:

$$\begin{aligned} H_j &= (D_z - J_{c1} + J_{c2})S^2\cos^2\alpha \\ &\quad + [J_{c1}\cos(2\pi q_{IC}) + J_{c2}\cos(4\pi q_{IC})]S^2\sin^2\alpha. \end{aligned} \tag{5}$$

Here note that $H_j$ is not $j$-dependent, therefore one can minimize the total Hamiltonian by minimizing $H_j$. Taking partial derivative of (5) with respect to $q_{IC}$ and $\alpha$, one gets

$$\frac{\partial H_j}{\partial q_{IC}} = -2\pi S^2\sin^2\alpha[J_{c1}\sin(2\pi q_{IC}) - 2J_{c2}\sin(4\pi q_{IC})], \tag{6}$$

$$\frac{\partial H_j}{\partial \alpha} = 2S^2\sin\alpha\cos\alpha[-D_z + J_{c1}(\cos(2\pi q_{IC}) + 1) + J_{c2}(\cos(4\pi q_{IC}) - 1)]. \tag{7}$$

From (6), we see that when the canting angle $\alpha$ is finite, the incommensurability $q_{IC}$ is not dependent on the canting angle, and is only a function of the $J_{c1}/J_{c2}$ ratio, as stated in the main text. For $q_{IC} = 0.46$, we have $J_{c1}/J_{c2} = 3.874$. In ref. 27, Beckman et al. deduce from susceptibility measurements that the $J_{c1} = 3.5$ meV and $J_{c2} = 0.9$ meV. These values are much different from exchange couplings $J_{c1}$ and $J_{c2}$ determined from the $c$-axis spin wave dispersion of FeGe (Fig. 2b).

However, from equation (7), we see that only by setting $\alpha = 0$ or 90°, one can achieve the lowest energy for the spin Hamiltonian. As a consequence, an $\alpha = 18°$ magnetic structure is prohibited in this model. To understand the observed incommensurate magnetic structure, the spin Hamiltonian must be adjusted. In ref. 27, Beckman et al. assumed a higher-order anisotropy $D_4 S_z^4$ term, which turns Eq. (7) into

$$\begin{aligned} \frac{\partial H_j}{\partial \alpha} &= 2S^2\sin\alpha\cos\alpha[-D_z - 2D_4 S^2\cos^2\alpha \\ &\quad + J_{c1}(\cos(2\pi q_{IC}) + 1) + J_{c2}(\cos(4\pi q_{IC}) - 1)]. \end{aligned} \tag{8}$$

Since the higher-order anisotropy put a term with $\alpha$ into the square bracket of equation (7), one can expect a canting angle that is not 0 or 90°. By setting the part in the square bracket equal to zero, putting together $J_{c1}/J_{c2} = 3.874$, $\alpha = 18°$, $q_{IC} = 0.46$, and $S = 1$, one will have

$$1.809 D_4 + D_z + 0.005123 J_{c1} = 0. \tag{9}$$

If $D_z$ and $J_{c1}$ are known, $D_4$ can be calculated accordingly. Note here only when $D_4 > 0$ (favoring an easy plane) will the Hamiltonian be convex with respect to $\alpha$. Therefore, it requires $|J_{c1}| < 195.2|D_z|$ for positive AFM $J_{c1}$ and negative $D_z$ favoring an easy axis.

The previous model gives a minimum parameter set necessary to induce an incommensurate canting phase. It is possible to have other exchange interactions, such as off-diagonal interactions as well as biquadratic interactions. Here we will give a more complete analysis of the possible exchange interactions: First, we consider bilinear exchange interactions. Intralayer interactions do not contribute to the [0, 0, $L$] spectrum, so we will only discuss interlayer exchanges. For off-diagonal interactions with $S_{i\alpha}J_{\alpha\beta}S_{j\beta}$, only when $\{\alpha, \beta\} \in \{x, y\}$ does the matrix element $J_{\alpha\beta}$ take effect in LSWT for collinear AFM magnetic structure, because any bilinear term containing only one $S_z$ will only have odd numbers of magnon operators and should be omitted in

LSWT. This gives the possible configuration $J_{\alpha\beta}$ as $\begin{bmatrix} A & D+E & 0 \\ D-E & B & 0 \\ 0 & 0 & C \end{bmatrix}$

Where $\{A, B, C\}$ is the anisotropic exchange, $D$ is the strength of symmetric off-diagonal exchange, and $E$ is the antisymmetric exchange, i.e., the DM interactions. In LSWT, the anisotropic exchange has the same effect as single-ion anisotropy when written in bilinear spin operators in $k$-space, and will only lift the whole spin wave spectra by a certain energy depending on the difference between $A$, $B$, and $C$, and will open a gap at the lowest energy. The symmetric off-diagonal exchange will induce the same effect. The DM interaction has no impact on the spin wave spectrum for pristine AFM FeGe as discussed in the main text. For multi-spin interactions, we consider biquadratic exchanges as an example. In the linear approximation, the biquadratic exchange produces the same spin waves, but the effective exchange coupling is modified and proportional to the temperature-dependent ordered spin $S^2$. If this temperature-dependent interaction is considered one of the origins of the incommensurability, then the incommensurability wavevector should also change as a function of temperature, which alternates the ordered spin. Figure 5f in the main text shows the incommensurate wavevector $q_{IC}$ varies from 0.455 to 0.465, meaning the effective $J_{c1}/J_{c2}$ between 3.83 and 3.91. If biquadratic interactions are the reason for the change of $q_{IC}$, the relative energy scale will not be larger than 3% of $J_{c1}$, and should not be the main reason for the incommensurability. This argument can also be used to exclude other multi-spin interactions, such as the three-spin interaction in introducing the incommensurate order. Therefore, the $J_{c1} - J_{c2}$ model is a minimum effective model for consistently explaining the incommensurability through the whole temperature range.

### Roles of the DM interaction and additional anisotropy from CDW

The DM interaction works in a similar way as the Heisenberg exchange. Using equation (4) to calculate the DM energy, we can get

$$H_j^{\text{DM}} = A_j^{\perp} S^2 \sin(2\pi q_{IC})\sin^2\alpha, \tag{10}$$

where $A_j^{\perp}$ is the net DM interaction between the $j$th and $(j+1)$th layers of atoms. In the A-type AFM phase, $A_j^{\perp}=0$ as shown in Fig. 2a. While the detailed crystalline structure of the CDW phase is unknown, a non-zero $A_j^{\perp}$ will be possible in the CDW phase. Adding equation (10) to equation (5), we can see the $H_j^{\text{DM}}$ adds up to the second term of the right part in equation (5), which does not change the fact that $\frac{\partial H_j}{\partial \alpha}$ can only achieve its lowest energy state at $\alpha=0$ or 90°. Although we have to note that in the local exchange picture, the change of DM interaction between AFM and CDW phase will alternate the incommensurate wavevector $q_{IC}$, the associated energy scale will be smaller than 3% of $J_{c1}$ using the same argument as in biquadratic interactions.

According to the recent X-ray diffraction experiments[35], one of the most prominent features of the CDW-induced lattice distortion is the movement of Fe and Ge atoms along the $c$-direction, suggesting a $c$-axis modulation of the Fe and Ge atoms. If this is the case, the Fe environment will not be mirror symmetric along the $c$-axis, and odd-parity anisotropy terms ($D_1 S_z$, $D_3 S_z^3$, etc.) will be present in the Hamiltonian. The detailed angle dependence of the magnetic anisotropy will require further neutron and magnetometry experiments to resolve.

Although our inelastic neutron scattering study of spin excitations in the main text eliminated the possibility that the local exchange interactions, including the DM interaction, can give rise to the incommensurate phase, the aforementioned theory is still valuable. Assuming that the incommensurate phase originates from Fermi surface nesting, one can write down a Landau theory with the in-plane moment as the order parameter, and it can generate a canting phase with a certain set of parameters. Even in this case, the exchange interactions and quadratic term of anisotropy will contribute to the

quadratic term of the Landau theory, and the higher-order anisotropy will affect its higher-order terms, effectively competing with the Fermi surface nesting and controlling the incommensurate order parameter.

### Magnetic intensities of the incommensurate phase

Figure S2 shows the detailed cuts of magnetic excitations shown in Figs. 2f and 4f of the main text. All the data are integrated according to the range specified in the main text. Figure S3 shows the temperature dependence of the incommensurate magnetic peaks under an 11-T in-plane field. While the incommensurate peak intensity reduces significantly as shown in Fig. 4c of the main text, its temperature dependence is not changed. The temperature dependence of the (0, 0, 0.5) peak, a combination of the CDW superlattice and AFM peak from the in-plane moment induced by the in-plane magnetic field, mostly follows the CDW temperature dependence (Fig. S3b). Figure S4 shows the spin excitations taken at 2 K, 70 K, and 120 K under an in-plane field of 2-T, not much different from the 0-T data.

## Data availability

The data that support the plots in this paper and other findings of this study are available from the corresponding author on reasonable request.

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

## Acknowledgements

The neutron scattering and single-crystal synthesis work at Rice was supported by US NSF-DMR-2100741 and by the Robert A. Welch Foundation under grant no. C-1839, respectively (P.D.). M.Y. acknowledges support by the U.S. DOE grant No. DE-SC0021421 and the Robert A. Welch Foundation, Grant No. C-2175. A portion of this research used resources at the Spallation Neutron Source, a DOE Office of Science User Facility operated by Oak Ridge National Laboratory. The access of Pelican instrument at ANSTO (P17255) is gratefully acknowledged. B.Y. acknowledges the financial support by the European Research Council

(ERC Consolidator Grant "NonlinearTopo", No. 815869) and the ISF - Personal Research Grant (No. 2932/21).

## Author contributions
P.D. and M.Y. conceived and managed the project. The single-crystal FeGe samples were grown by X.T. and B.G. Neutron scattering experiments were carried out by L.C., X.T., B.W., G.G., F.Y., D.H.Y., R.A.M., and analyzed by L.C. DFT calculation is carried out by H.T. and B.Y. The paper was written by L.C. and P.D. with inputs from all coauthors.

## Competing interests
The authors declare no competing interests.
