## [Peer Review File · Nature Communications]

Reviewers' Comments:

Reviewer #1:

Remarks to the Author:

I consider that the authors have responded to the referees' comments, and the current version of the manuscript is an improvement compared to the first version I have seen previously. I therefore recommend the paper for publication in *Nature Communications* without further review.

Just two remarks that the authors should still consider:

1. I saw a new preprint by some of the same authors on arXiv:2309.14314, and I noticed an inconsistency in the values of transition temperatures reported in these two papers. For example, T_{CDW} is reported as 100 K here and as 110 K in the other paper. This is not so important, as the values are anyway reported as approximate, but I see no reason not to unify them among the two papers. The data in both papers clearly indicate that the CDW transition happens well above 100 K, closer to 110 K.

2. With the application of magnetic field, the spin gap opens as seen in Fig. 3g. Is it known where the corresponding spectral weight goes? Is it transferred to energies right above the gap or to higher energies? The same question applies to the temperature dependence of the spin gap. If it is reduced on cooling, does the corresponding spectral weight come from higher energies, from the reduction of magnetic Bragg intensity, or is somehow redistributed between the commensurate and incommensurate components? It would be helpful to add a few sentences in the text to clarify this point.

Reviewer #2:

Remarks to the Author:

The authors have substantially revised the manuscript, providing additional explanations and introducing further clarifications. I remain satisfied that the results are interesting and that they are appropriate for *Nature Communications*. The key observation is that the temperature dependence of the low energy incommensurate excitations is not the same as that of the higher energy commensurate excitations (which follow the temperature dependence expected for bosonic modes).

It is interesting that the structure of the magnetic excitation spectrum is somewhat similar to the hourglass spectrum exhibited by hole-doped cuprate superconductors and some cobaltates, which are mentioned in the introduction. In the case of the cuprates and cobaltates, the hourglass structure has been explained by the existence of incommensurate charge stripe order. Have the authors considered whether there might be a CDW modulation along the c -axis that might be associated with the incommensurate magnetic order and low energy dynamics? Can such a CDW be ruled out from observation?

My only other comments are with respect to the evidence for fermi surface nesting from the DFT calculations (Fig. S6 c,d and Fig. R1). In point #3 of my previous report I asked what is being plotted in Figs. 1h-j. The authors have not answered my question directly. Are these figures showing slices through $\chi(q)$ after averaging over some interval in k_y ? Please clarify in the figure caption. Also, the vertical axis of the new figures S6 c,d in the Supplementary Material are labelled "Intensity". What does this mean? Is it actually $\chi(q)$? Again, please be more specific in the caption.

Finally, the peaks labelled q_{IC} in Fig. S6 c,d and Fig. R1 are not particularly prominent, and indeed there are larger peaks at other wave vectors. This is surprising because if fermi surface

nesting is the mechanism for the incommensurate magnetic correlations, one might expect the peaks in $\chi(q)$ at $q = q_{IC}$ to be the strongest peaks. Otherwise the system does not save much energy by forming IC order at q_{IC} . What is the reason why the IC order does not form at one of the other wave vectors where there are prominent peaks in $\chi(q)$? There should be a balanced and more complete discussion about the interpretation of the calculated $\chi(q)$ in terms of the fermi surface nesting scenario for IC magnetic ordering.

REVIEWER COMMENTS

Reviewer #2 (Remarks to the Author):

I consider that the authors have responded to the referees' comments, and the current version of the manuscript is an improvement compared to the first version I have seen previously. I therefore recommend the paper for publication in Nature Communications without further review.

Thank you very much for recognizing the improvements made to the manuscript. We sincerely appreciate your recommendation for publication in Nature Communications.

Just two remarks that the authors should still consider:

1. I saw a new preprint by some of the same authors on arXiv:2309.14314, and I noticed an inconsistency in the values of transition temperatures reported in these two papers. For example, T_{CDW} is reported as 100 K here and as 110 K in the other paper. This is not so important, as the values are anyway reported as approximate, but I see no reason not to unify them among the two papers. The data in both papers clearly indicate that the CDW transition happens well above 100 K, closer to 110 K.

Thank you for pointing out the inconsistency between the two papers. We have revised the manuscript to ensure that the value of T_{CDW} is consistently reported as 110 K.

2. With the application of magnetic field, the spin gap opens as seen in Fig. 3g. Is it known where the corresponding spectral weight goes? Is it transferred to energies right above the gap or to higher energies? The same question applies to the temperature dependence of the spin gap. If it is reduced on cooling, does the corresponding spectral weight come from higher energies, from the reduction of magnetic Bragg intensity, or is somehow redistributed between the commensurate and incommensurate components? It would be helpful to add a few sentences in the text to clarify this point.

The magnetic field increases the effective magnetic anisotropy. Therefore, the overall spin wave dispersion will be lifted to a higher energy, and consequently the spectral weight is also lifted. The same scenario happens for temperature dependence. As temperature decreases, the magnetic anisotropy drops (for some unknown reason), and the whole spin wave spectrum is modified to a lower energy, and we can effectively say that the spectral weight comes from higher energy. We have added a few sentences for clarification.

Reviewer #3 (Remarks to the Author):

The authors have substantially revised the manuscript, providing additional explanations and introducing further clarifications. I remain satisfied that the results are interesting and that they are appropriate for Nature Communications. The key observation is that the temperature dependence of the low energy incommensurate excitations is not the same as that of the higher energy commensurate excitations

(which follow the temperature dependence expected for bosonic modes).

We appreciate the referee's time and efforts in reviewing our manuscript. We are glad to hear that the revisions we introduced were satisfactory and that the referee finds the results interesting and appropriate for Nature Communications.

It is interesting that the structure of the magnetic excitation spectrum is somewhat similar to the hourglass spectrum exhibited by hole-doped cuprate superconductors and some cobaltates, which are mentioned in the introduction. In the case of the cuprates and cobaltates, the hourglass structure has been explained by the existence of incommensurate charge stripe order. Have the authors considered whether there might be a CDW modulation along the c-axis that might be associated with the incommensurate magnetic order and low energy dynamics? Can such a CDW be ruled out from observation?

We thank the referee for raising this possibility on the origin of the IC order. In the neutron diffraction experiment of the IC peaks, we can see that the IC intensity follows the magnetic form factor of Fe atom, indicating the IC peaks are purely magnetic (Extended data fig.8d,8e in Nature 609, 490–495 (2022)). Therefore, we did not observe definite signatures of CDW modulating the underlying lattice. We have also carried out neutron polarization analysis (unpublished), and these results show conclusively that IC peaks are purely magnetic without structural component. We are in the process of preparing neutron polarization analysis data for publication. Yes, we can rule out IC CDW. These results are also in agreement with CDW seen by X-ray scattering experiments (H. Miao et al. Ref. [34], now published in Nature Communications).

My only other comments are with respect to the evidence for fermi surface nesting from the DFT calculations (Fig. S6 c,d and Fig. R1). In point #3 of my previous report I asked what is being plotted in Figs. 1h-j. The authors have not answered my question directly. Are these figures showing slices through $\chi(q)$ after averaging over some interval in k_y ? Please clarify in the figure caption. Also, the vertical axis of the new figures S6 c,d in the Supplementary Material are labelled "Intensity". What does this mean? Is it actually $\chi(q)$? Again, please be more specific in the caption.

We thank the referee for asking for clarification. In Fig.1h-j, we are plotting the spin-resolved band structure in the shaded plane in fig.1d. These plots are not averaged in k_y . The label 'Intensity' in fig.S6 is actually $\chi(q)$. These corrections are made in the revised draft.

Finally, the peaks labelled q_{IC} in Fig. S6 c,d and Fig. R1 are not particularly prominent, and indeed there are larger peaks at other wave vectors. This is surprising because if fermi surface nesting is the mechanism for the incommensurate magnetic correlations, one might expect the peaks in $\chi(q)$ at $q = q_{IC}$ to be the strongest peaks. Otherwise the system does not save much energy by forming IC order at q_{IC} . What is the reason why the IC order does not form at one of the other wave vectors where there are prominent peaks in $\chi(q)$? There should be a balanced and more complete discussion about the interpretation of the calculated $\chi(q)$ in terms of the fermi surface nesting scenario for IC magnetic ordering.

We agree with the referee that the peak in $\chi(q)$ in those figures are not too prominent. In the DFT calculation in the FM phase, the $\chi(q)$ at $q=0.35$ is indeed stronger than that at $q=0.46$, however, the AFM phase calculation indeed gives strongest $\chi(q)$ peak at q_{IC} (the peak at $q=0.5$ corresponds to the FBZ folding for the AFM phase). Additionally, the fact that the CDW phase modifies the IC excitation intensity suggests that the Fermi surface modulation brought by the CDW, mainly associated with the Van Hove singularities, is crucial to the formation of the IC phase, which is not reflected in the DFT calculations. We have added some discussions on this issue.